# Research on Users' Privacy-Sharing Intentions in the Health Data Tracking System Providing Personalized Services and Public Services

Shugang Li , Kexin Peng , Boyi Zhu, Ziyi Li, Beiyan Zhang, Hui Chen * and Ruoxuan Li *

School of Management, Shanghai University, Shanghai 200444, China; westside_li@163.com (S.L.); 1023926458@shu.edu.cn (K.P.); boyi_star@163.com (B.Z.); lt122500@163.com (Z.L.); zhangbeiyan@shu.edu.cn (B.Z.)
* Correspondence: chenhui@shu.edu.cn (H.C.); rachel-li@shu.edu.cn (R.L.)

**Abstract:** The utilization of user privacy data in the health data tracking system (HDTS) offers numerous benefits for businesses and public services, contingent upon users' privacy sharing intentions. However, previous research neglected users' preferences for public services and focused mainly on personalized services. Additionally, traditional privacy calculus theory has a limited focus on data security, leaving gaps in understanding individual and societal aspects. This study aims to fill these gaps by examining the influence of risk perception and factors like potential loss expectations, perceived personalized service benefits, group value identification, perceived public service utility, and perceived privacy on privacy sharing intentions in the context of personalized and public services. The results indicate a positive relationship between individual privacy protection perception and data sharing intention, as well as a positive relationship between group value identification and perceived public service utility with individuals' privacy sharing intentions. Moreover, this research uncovers the moderating effect of information type sensitivity on the impact of perceived privacy and perceived public service utility on privacy sharing intentions, while there is no moderating effect of information type sensitivity on the relationship between group value identification and privacy sharing intentions. We recommend improving individual privacy education, ensuring data use transparency, and fostering identification with common group values to increase users' privacy sharing intentions.

**Keywords:** privacy calculus; privacy sharing intentions; group value identification; perceived public service utility; perceived privacy



## 1. Introduction

Health data tracking systems are systems that monitor and track individuals' health status, behavior patterns, and health risks via the collection and analysis of their health data [1]. Health data tracking systems help individuals understand and manage their health conditions, providing personalized health advice and interventions. By collecting and analyzing large-scale health data, these systems can generate insights on health trends, risk warnings, and personalized recommendations, delivering relevant health information and advice to users [2].

Currently, health data tracking systems have drawn interest from companies worldwide, with some well-known companies launching their own systems, such as Apple Health, Google Fit, Fitbit, Samsung Health, etc.

In health data tracking systems, the use of users' private data offers several benefits for businesses. Firstly, by analyzing users' health data, companies can gain insights into users' needs and behavior patterns, optimizing products and services to provide more personalized and valuable health solutions [3]. Secondly, the collection and analysis of large-scale user data support innovation and research activities for the development of

new products and services to meet market demands [4]. Lastly, user privacy data can help companies identify market opportunities and insights into different demographic groups' health needs and preferences, enabling the formulation of corresponding marketing strategies and product positioning and enhancing market competitiveness and business success [5].

In addition to benefiting businesses, the use of user privacy data also has positive implications for public services [6]. Companies can share users' health data with public health departments, medical institutions, and other partners, supporting the implementation and improvement in public services. Via data sharing with relevant institutions and partners, companies can engage in collaborations concerning public health policy-making, epidemic control, and disease prevention, collectively working toward improving public health [7].

The utilization of user privacy data in businesses and public services offers several advantages. However, the realization of these benefits is contingent upon users' willingness to share their data. Previous research has mainly focused on users' willingness to share data for personalized services [8,9], neglecting their preferences concerning public services. This study aims to address this gap by simultaneously considering both personalized and public services, providing a more comprehensive approach to the system. Moreover, traditional research has often disregarded users' awareness and participation in public services and social benefits, but this study seeks to balance individual and collective interests to gain a better understanding of users' privacy sharing intentions.

Researching users' demand for high-quality health data tracking services and their privacy sharing intentions presents significant challenges. Understanding users' values, perceptions of personal privacy, and their considerations when balancing privacy with system services and social benefits is crucial [10]. Additionally, studying users' risk perception and trust-building during data sharing, including concerns about data misuse and privacy breaches, is essential [11]. Lastly, exploring how users weigh personalized service demands against public service contributions is vital. This involves a deep understanding of user behavior, psychology, and social factors [12]. The objective of this study is to provide valuable guidance for the development and implementation of health data tracking systems, taking into account the interests and risk factors from the perspectives of individuals, enterprises, and society.

To achieve our research objectives, we aim to delve into the dynamics and variations of privacy sharing decisions and establish a model to understand the influencing mechanisms of users' privacy sharing intentions [13]. We will comprehensively consider the needs of individuals, enterprises, and society, striving to better understand users' privacy sharing intentions. In-depth research on user behaviors and psychological factors will provide comprehensive guidance for health data tracking systems. Our goal is to strike a balance between personalized services and public services, ultimately offering valuable references for future system design and management.

Privacy calculus is a computational approach that safeguards individual privacy via data encryption and privacy maintenance during data computation, facilitating data sharing and analysis [14]. By embracing privacy calculus theory, we can gain a comprehensive framework to explore the delicate balance of interests, risks, dynamics, and variations in users' decisions regarding privacy sharing [15]. Privacy calculus theory helps unveil behavioral patterns and factors influencing privacy sharing choices, deepening our understanding of how individuals navigate between personal privacy protection and data sharing. Moreover, privacy calculus theory offers essential guidance for designing privacy calculus systems and services, ensuring effective protection of individual privacy while achieving data sharing and analysis goals [14]. Multiple studies have expanded the reach of privacy calculus theory, incorporating trust, social influence, and cognitive factors related to personal data management, strengthening its predictive capabilities. For instance, research has investigated users' trust in service providers as a significant predictor of privacy-related intentions alongside risk–benefit assessments [16,17]. Additionally, studies have explored the impact of social influence, benefits, and privacy concerns, especially in the context

of COVID-19 contact tracing app adoption [18]. Integrating privacy self-efficacy into the model has explained user behaviors on platforms like Facebook [19].

However, traditional privacy calculus theory has primarily focused on data security and privacy protection, leaving gaps in understanding individual and societal aspects. In particular, there is a lack of in-depth research concerning personalized services, social identity, and public interests, all of which play vital roles in shaping users' perceptions and cognition related to personalized services, social values, and public interests. To advance privacy calculus models, it is crucial to integrate individual and societal factors [20]. Research methods that comprehensively consider these factors represent innovative steps in advancing privacy calculus theory and its practical applications. By addressing these gaps, privacy calculus can better balance data security with factors like personalized services, social identity, and public interests, offering more comprehensive and well-rounded solutions for privacy protection and data sharing.

This study first investigates individual risk perception and explores how perceived potential loss expectations and benefits of personalized services impact users' perceived privacy to bridge the gap in traditional privacy calculus theory [21]. Additionally, users' perceptions of privacy, group value identification, and perceived public service utility play a significant role in shaping their privacy sharing intentions. Investigating the relationship between perceived privacy, group value identification, perceived public service utility, and privacy sharing intentions can fill the gap in privacy calculus theory concerning users' privacy sharing intentions and offer critical insights to understand and guide users' privacy sharing behaviors from the perspective of social factors.

Furthermore, this study investigates the dynamics and variations in individuals' privacy sharing decisions, considering the moderating effects of information type sensitivity on the relationship between group value identification, perceived public service utility, perceived privacy, and privacy sharing intentions. These findings address aspects that traditional privacy calculus theory has not adequately explored and are crucial for achieving a balance between individual privacy protection and public services.

We made the following discoveries: (1) Consumers will experience a psychological state of perceived privacy when facing the risks and benefits brought by personalized service; more specifically, consumers' potential loss expectations have a negative impact on perceived privacy while personalized services have positive impact on perceived privacy. This fills the gap in how users balance risks and benefits in the decision-making process. (2) The positive impact of privacy perception on privacy sharing intention indicates that users' perception of privacy protection can directly affect their willingness to share data. Users realize that their privacy is respected, which can inspire them to participate more actively in data sharing and provide more control options. (3) We conducted separate investigations on the positive relationships between group value identification, public service utility perception, and individual privacy sharing intentions. These findings establish a fundamental basis for addressing the influence of social factors on privacy sharing. (4) The information type sensitivity has a positive moderating effect on the impact of perceived privacy on privacy sharing intentions, while it has a negative moderating effect on the impact of perceived public service utility on privacy sharing intentions. This contributes to a better understanding of the relationship between individual privacy protection behavior and information sensitivity.

Thus, to enhance the balance between privacy protection and personalized services, we recommend that health data tracking system designers prioritize the privacy-service balance in system design to ease privacy concerns and enhance the combination of personalized and public services. It is also significant to promote individual privacy education and transparency in data used to empower individuals to make informed decisions regarding privacy protection. Additionally, users' group values identification and public service value perception should be strengthened to boost data sharing intent. Finally, personalized privacy measures could be developed to encourage data sharing.

The rest of this study is organized as follows. The literature review on privacy calculus theory is presented in Section 2. The research models and hypotheses are described in Section 3. The research methodology is shown in Section 4. The results of the experiments are provided in Section 5. Finally, Section 6 concludes this study.

## 2. Literature Review

Privacy calculus theory is a widely accepted framework for understanding privacy-related behaviors and intentions, revolving around two central concepts: risks and benefits [22,23]. Risks pertain to individuals' belief in the potential losses resulting from information disclosure, representing inhibitions toward sharing information [23]. Privacy concerns have been used in some studies to measure the level of risk associated with information disclosure [24–26]. On the other hand, benefits encompass the gains individuals anticipate from sharing information, encouraging disclosure [23]. The specific forms of benefits vary depending on the context of information disclosure. For instance, in social networks, benefits often include establishing social relationships, obtaining social emotional support, and expressing oneself [26–29]. In the context of mobile applications, benefits arise from users sharing personal information like location and preferences, resulting in enhanced app usefulness and user experience [30].

According to privacy calculus theory, when individuals make decisions related to privacy, they engage in a process of privacy calculus, wherein they weigh the expected benefits against the potential risks of information disclosure. Personal intentions and subsequent behaviors are positively influenced by perceived benefits and negatively affected by potential risks.

While privacy calculus theory has been extensively applied to explain various behaviors such as system usage [25,31], social media friending [29], information disclosure [21,26,28,32], false statements [32], and information withdrawal [19], it mainly focuses on the trade-offs between risks and benefits in individual privacy-related decisions. However, behavioral decision-making is often more intricate, and outcomes can be influenced by other critical factors. Some studies have extended privacy calculus theory by incorporating trust factors, social influence factors, and cognitive factors related to the ability to protect and manage personal information, enhancing its predictive and explanatory power in specific contexts. For instance, Bol et al. (2018) found that personalization has minor trust and benefit impacts, with variations across contexts such as news and commerce [17]. Dienlin and Metzger (2016) confirmed both privacy concerns and privacy self-efficacy can positively predict the use of self-withdrawal [19]. Fox et al. (2021) noted citizens' initial acceptance depends on health benefits and social influence, while reciprocity and privacy concerns have lasting and limited impacts [18]. Kim and Kim (2020) discovered that users' intention to share personal information has a minor effect on their actual disclosure behavior. In addition, control over personal information boosts users' trust in social networking service providers and positively influences intention and behavior while reducing privacy concerns [27]. Leon et al. (2021) revealed that higher perceived privacy risk reduces the intention to adopt drone delivery, which is influenced by privacy disposition, concerns, and legislation. Perceived usefulness is the key adoption factor, mitigating privacy risk when it is deemed useful. Trust also affects the intent to use drone delivery [16].

Despite the value of privacy calculus theory, there are some research gaps that necessitate further exploration and refinement. Firstly, more in-depth research is needed to understand the connection between personalized service demands and individuals' willingness to share personal information [33]. With the increasing prevalence of personalized services, people's desires and willingness to protect their privacy are constantly evolving, requiring more research to balance these demands and provide acceptable privacy computing solutions.

Secondly, the theory should be extended to account for social and group factors. Privacy protection is not solely an individual concern; it also involves societal and group-level needs and interests [20]. Therefore, further research is necessary to comprehensively

consider social and group factors, find a balance between individual privacy protection and societal benefits, and formulate corresponding privacy computing strategies to cater to different groups' needs.

Furthermore, the theory's applicability to public needs and social interests requires improvement. Privacy protection is closely related to public needs and social interests, such as medical research and social surveys [34,35]. Therefore, more exploration is needed to strike a balance between individual privacy and public needs, offering more comprehensive and sustainable privacy computing solutions.

Lastly, the dynamic and individual variability in privacy sharing decisions is an aspect that traditional privacy calculus theory has not fully addressed. Individuals' willingness and behaviors regarding privacy sharing are often subject to dynamic changes [36], and different types of sensitive information may lead to distinct responses [37]. Hence, further research is necessary to understand the dynamics and individual variability in privacy sharing decisions, better addressing the balance between personal privacy protection and the development of public services.

To address these research gaps and advance the field of privacy calculus, this study investigates the link between personalized service demands and privacy sharing intentions in the context of health data tracking systems, as well as social and group factors, public needs and social interests, and the dynamic and individual variability in privacy sharing decisions. We propose a novel privacy computing model to provide enhanced privacy protection theories and practices.

## 3. Research Model and Hypotheses

### 3.1. Potential Loss Expectations, Perceived Personalized Service Benefits, Perceived Privacy, and Privacy Sharing Intentions

The privacy sharing intention refers to the extent to which individuals are willing to disclose personal information to others, which, in the context of health data tracking systems, manifests as whether individuals are willing to share their health data with platforms, government agencies, and others.

According to privacy calculus theory, individuals engage in a risk–benefit trade-off before disclosing personal information [21,38,39]. Perceived potential risks decrease the willingness to disclose information, while perceived potential benefits increase the willingness to disclose [17,28]. Currently, most research focuses on the direct causal relationship between risk perception, benefit perception, and information disclosure intention [40–44].

Perceived privacy refers to individuals' cognition and attitudes regarding the collection, use, and sharing of their information [45]. In health data tracking systems, perceived privacy reflects the degree of concern and apprehension individuals have regarding the privacy of their health data.

Potential loss expectations refer to individuals' expectations of future losses or risks [46]. This includes the risk of personal data misuse, leakage, or unauthorized use. According to privacy calculus theory, individuals' concern for the privacy of personal information is reasonable, as they may be worried about potential losses resulting from data misuse, leakage, or misappropriation [47]. Privacy calculus theory emphasizes individuals' concerns about whether the sharing of personal information aligns with their expectations and needs, highlighting their control and autonomy over their personal information [48]. Currently, there is a lack of research that incorporates the relationship between potential loss expectations, personalized service benefits, and perceived privacy into privacy calculus theory.

Health data tracking systems that provide personalized and public services typically require the collection and processing of users' sensitive health data. Users' awareness of their right to know and control their data is a crucial factor in privacy protection [49]. When users realize that potential losses may occur, they may feel a reduction in control over their data, which can impact perceived privacy [50]. Additionally, users may be concerned about information leakage, unauthorized access, and misuse of data, as well as the trade-offs

between personalized data use and public benefits, which may reduce their trust in the system's privacy protection capabilities, thereby affecting the degree of perceived privacy. Based on this, we propose the following hypothesis:

**H1.** *In health data tracking systems that provide personalized and public services, users' potential loss expectations negatively affect perceived privacy.*

The realization of personalized services typically requires the collection and analysis of users' health data to provide personalized recommendations or optimize service experiences [51]. Users' awareness that personalized services can better meet their preferences and needs enhances their sense of control and autonomy over the use of their data [52]. This perception is critical for personal privacy protection as it increases individuals' understanding and participation in the use of their data, thereby enhancing perceived privacy [53].

Moreover, the benefits of personalized services also include providing personalized medical resources and enhancing public services [54]. By tracking and analyzing a large amount of health data, the system can better identify public health trends, predict disease outbreaks, and improve public health strategies [55]. Users' realization that their personal health data can not only provide personalized services for themselves but also contribute to societal health may increase their willingness to actively share their health data to promote better medical and public health services. This sense of shared benefit can positively impact users' perceived privacy.

Therefore, we propose the following hypothesis:

**H2.** *In health data tracking systems that provide personalized and public services, users' perceived personalized service benefits positively affect perceived privacy.*

Traditional privacy calculus theory mainly focuses on the development of data encryption and privacy protection technologies, often treating individuals as passive objects of privacy protection [56]. It overlooks the importance of individuals' willingness and participation in privacy protection. Additionally, privacy calculus also neglects to seek a balance between individual privacy protection and societal interests. To fill the existing research gap, this study explores the relationship between users' perceived privacy and privacy sharing intentions in health data tracking systems that provide personalized and public services.

When users perceive that the system values privacy protection and has effective measures in place, they may trust the system more and develop a positive attitude toward their privacy sharing intentions [57]. Additionally, health data tracking systems that provide personalized and public services can reveal insights and patterns in personal health and public health via data analysis and mining. When users realize that their privacy sharing can bring mutual benefits to themselves and society, they may be more willing to proactively share their health data, leading to a positive impact on their privacy sharing intentions. This sense of shared benefit is crucial as it increases users' understanding and engagement in the use of their data, thereby enhancing perceived privacy.

Based on this, we propose the following hypothesis:

**H3.** *In health data tracking systems that provide personalized and public services, users' perceived privacy positively affects their privacy sharing intentions.*

*3.2. Group Value Identification and Privacy Sharing Intentions*

Group value identification (GVI) is defined as the degree to which an individual believes that people who are important to them think they should perform a certain behavior [58]. Group value identification reflects the perceived social pressure when individuals decide whether to perform a certain behavior. However, traditional privacy calculus theory overlooks the inclusion of the relationship between group value identification and

individuals' privacy sharing intentions. In fact, when individuals believe that important groups endorse their privacy sharing behavior, it can lead to social influence, trust, support, and social identity [59]. The pursuit of belongingness and social identity in society is a fundamental need for individuals [60]. When individuals identify with the values of their social groups and these groups endorse privacy sharing as an important behavior, individuals are more likely to engage in such behavior, i.e., sharing health data. Moreover, group expectations and viewpoints play a crucial role in individual behavior. When individuals perceive that important members of their social groups believe that privacy sharing is the right behavior, they face social pressure from the group to perform this behavior, thereby increasing their privacy sharing intentions. Thus, this study proposes the following hypothesis:

**H4.** *In health data tracking systems that provide personalized and public services, group value identification positively affects privacy sharing intentions.*

### 3.3. Perceived Public Service Utility and Privacy Sharing Intentions

Perceived public service utility refers to the degree to which an individual perceives the utility of the public services provided by the health data tracking system [61]. However, specific research and discussions regarding perceived public service utility in traditional privacy calculus theory are still lacking. It omits the consideration of perceived public service utility as an important factor influencing individuals' privacy sharing intentions [42]. Moreover, in the motivations and considerations of individual privacy sharing decisions, besides the existing motivations of maximizing benefits and personalized services, the introduction of perceived public service utility provides a new motivation that has been overlooked. In existing privacy calculus theory, individuals' privacy sharing intentions are usually studied and discussed by providing personalized services [17], with less consideration given to factors related to public services. Therefore, this study emphasizes the joint effect of personalized and public services and analyzes the impact of individuals' perceived public service utility on their privacy sharing intentions. When individuals perceive that using the health data tracking system and sharing personal data have a positive impact on public services, they may develop a sense of shared benefits [62]. Individuals may recognize that using the health data tracking system and sharing personal data can contribute to public services such as health monitoring and epidemic control, thus increasing their intentions to share information. Furthermore, when individuals perceive that using the health data tracking system and sharing personal data have a positive impact on public services, they may feel a sense of social responsibility [63]. Individuals may believe that they have a responsibility to contribute to public services. When individuals realize that using the health data tracking system and sharing personal data can help provide public services, they may be more willing to fulfill this social responsibility, thus increasing their intentions to share information.

**H5.** *In health data tracking systems that provide personalized and public services, perceived public service utility positively affects privacy sharing intentions.*

### 3.4. Moderating Effect of Information Type Sensitivity

Existing research has shown that individuals may establish different boundaries around different types of information collected to be disclosed and apply varied rules accordingly [64,65]. Some researchers have verified the direct or indirect effect of information type sensitivity on privacy sharing intentions. For instance, Malhotra et al. (2004) found that the type of information collected affects risk beliefs, trust beliefs, and information disclosure intentions [11]. In an experiment investigating consumers' information disclosure intentions in a smart income application, Kehr et al. (2015) found that the information type indirectly affects consumers' information disclosure intention by affecting perceived risks and perceived benefits [66].

According to sensitivity, we divide user privacy information into two types: high-sensitivity information (such as medical and health information, biometric information, and location information) and low-sensitivity information (such as health habits, health indicators, and health records) [67]. However, currently, there is a lack of research on how information type sensitivity may influence individuals' privacy sharing intentions in health data tracking systems that provide personalized and public services. Additionally, the dynamic and individual differences in individuals' privacy sharing decisions are aspects that traditional privacy calculus theory has not accurately revealed. By exploring the moderating effect of information type sensitivity in health data tracking systems that provide personalized and public services, this study fills the research gaps in these aspects of privacy calculus theory. It has important theoretical and practical significance for a deeper understanding of individual privacy protection behaviors and the balance between public interests and privacy rights.

When individuals perceive that using the health data tracking system may expose more sensitive personal information, their concerns and apprehensions about privacy may increase [68]. If individuals are more sensitive to information types, they may be more cautious and less willing to share privacy [21]. Therefore, information type sensitivity positively moderates the impact of perceived privacy on privacy sharing intentions.

When individuals are more sensitive to information types, they may pay more attention to personal privacy protection and maintain a higher level of identification with their rights and personal space [69]. Therefore, information type sensitivity weakens the influence of group value identification on privacy sharing intentions, making individuals less willing to share privacy. Thus, information type sensitivity may moderate the impact of group value identification on privacy sharing intentions.

When individuals are more sensitive to information types, they may pay more attention to personal privacy protection and weigh personal interests and public benefits more cautiously [15]. Therefore, information type sensitivity weakens the positive impact of perceived public service utility, making individuals more cautious in considering privacy sharing decisions. Thus, information type sensitivity may moderate the impact of perceived public service utility on privacy sharing intentions.

In summary, this study proposes the following hypotheses:

**H6.** *In health data tracking systems that provide personalized and public services, information type sensitivity positively moderates the impact of perceived privacy on privacy sharing intentions.*

**H7.** *In health data tracking systems that provide personalized and public services, information type sensitivity negatively moderates the impact of group value identification on privacy sharing intentions.*

**H8.** *In health data tracking systems that provide personalized and public services, information type sensitivity negatively moderates the impact of perceived public service utility on privacy sharing intentions.*

### 3.5. Control Variable

In addition to the factors mentioned above, other factors may also affect users' privacy sharing intentions. According to existing related research, we add eight control variables into the research model, including four demographic variables: gender [70], age [11], education [65], and income [65]; two variables related to respondents' personality: trust propensity, and altruism [65]; and two variables related to the respondents' experience: the frequency of respondents' privacy violations in the past (past privacy violations) [11] and whether respondents have experienced public health emergencies, for example, COVID-19 public health emergency.

The research model proposed in this study is shown in Figure 1.

**Figure 1.** Research model.

## 4. Research Methodology

### 4.1. Measurement Development

This study designed a questionnaire containing a scenario description [71], which consists of three parts. The first part investigates the basic information of the respondents, including gender, age, educational background, income, experience of public health emergencies, past privacy violations, media disclosure, trust tendency, and altruism. Among them, trust tendency and altruism are measured with the 7-point Likert scale, ranging from "1 = very disagree" to "7 = very agree".

The second part is the scenario description. In order to control the impact of information type sensitivity on the responses of respondents, this study set up a scenario description in the questionnaire. We created two types of scenarios based on the sensitivity of the collected privacy information. The questionnaire for the Class A scenario requires the respondents to provide the personalized and public services of the health data tracking system with high-sensitivity information. The information collected in the questionnaire for the Class B scenario involves low-sensitivity information. In the process of data collection, it is necessary to ensure that each respondent can only fill in the questionnaire of one scenario.

The third part is the measurement of scenario-related variables, including six constructs: perceived personalized service benefits, potential loss expectations, perceived privacy, group value identification, perceived public service utility, and privacy sharing intentions. The six variables, as well as the two control variables of trust propensity and altruism, are measured by the Likert seven scale, ranging from "1 = very disagree" to "7 = very agree". To ensure the reliability and validity of the scale, the items used for each construct are from the maturity scale in related research. Meanwhile, considering the context of personalized and public services of the health data tracking system, we slightly modified some items to adapt to the characteristics of personalized and public services of the health data tracking system context. To ensure the effectiveness of the content, we invited a group of experts (including a professor and two research assistants) to review the scale and revise and improve the semantics, coherence, and readability of the scale. The final measurement scales of the above eight variables are shown in Table 1.

**Table 1.** Construct measurement.

| Construct | Item # | Measurement Item | References |
|---|---|---|---|
| Trust propensity (TP) | TP1 | I usually trust people until they give me a reason not to trust them. | [65] |
| | TP2 | I usually give people the benefit of the doubt. | |
| | TP3 | My general approach is to trust new acquaintances until they prove I should not trust them. | |
| Altruism (AL) | AL1 | Helping others is one of the most important aspects of life. | [65] |
| | AL2 | I enjoy working for the welfare of others. | |
| | AL3 | My family and I tend to do our best to help those unfortunate people. | |
| | AL4 | I agree with the old saying, "It is better to give than to receive". | |
| Perceived personalized service benefits (PPSB) | PPSB1 | Providing the information to the system can make me more secure. | [45] |
| | PPSB2 | Providing the information to the system can make my life more convenient. | |
| | PPSB3 | In general, I think it is beneficial for me to provide the information to the system. | |
| Potential loss expectations (PLE) | PLE1 | Providing the information to the enterprises and government would involve many unexpected problems. | [45] |
| | PLE2 | It would be risky to provide the privacy information to the enterprises and government. | |
| | PLE3 | The potential for loss in providing the privacy information to the enterprises and government would be high. | |
| Perceived privacy (PP) | PP1 | I think I would have enough privacy when the privacy information is collected and used. | [72] |
| | PP2 | I think I would be satisfied with the privacy I have when the privacy information is collected and used. | |
| | PP3 | I think my privacy would be protected when the privacy information is collected and used. | |
| Group value identification (GVI) | GVI1 | Those people who are important to me would support me to provide the information to the system. | [58,73] |
| | GVI2 | People whose opinions I value would prefer me to provide the information to the system. | |
| Perceived public service utility (PPSU) | PPSU1 | This public service provided by the health data tracking system would be useful for personalized and public services. | [61] |
| | PPSU2 | This public service provided by the health data tracking system would enable the government to prevent and control the epidemic. | |
| | PPSU3 | This public service provided by the health data tracking system would enhance the effectiveness of personalized and public services. | |
| Privacy sharing intentions (PSI) | PSI1 | I am likely to provide the privacy to the system. | [21] |
| | PSI2 | It is probable that I will provide the privacy to the system. | |
| | PSI3 | I am willing to provide the privacy to the system. | |

### 4.2. Sample and Data Collection

A total of 60 sample data were collected in the pre-survey, and the analysis results of the pre-survey data show that the reliability and validity of the questionnaire data are good. According to the respondents' feedback information, this study further optimized and adjusted the questions, semantics, and structure of the questionnaire. For example, PR1 changed from "Providing the privacy information to the system would involve many intractable problems" to "Providing the privacy information to the system would involve many unexpected problems", making the items easier to understand.

From February 2021, this study distributed electronic questionnaires to users in Shanghai. By March 2021, a total of 255 samples were obtained. To ensure authenticity and reliability, we checked and screened the recovery samples. After eliminating the invalid questionnaires such as suspected repeated filling (the same IP address) and careless filling (more than 80% of the questions choose the same option), we obtained 232 valid samples, consisting of 136 scenario A and 96 scenario B respondents, and the effective rate

was 90.98%. There are no significant differences in gender, age, income, and education between the two types of data (scenario A and scenario B). Table 2 shows the demographic characteristics of the samples.

**Table 2.** Demographics for sample.

| Construct | Item # | Count | Percentage (%) |
|---|---|---|---|
| Gender | Male | 71 | 30.6 |
| | Female | 161 | 69.4 |
| | Less than 18 | 1 | 0.4 |
| Age | 18–30 | 207 | 89.2 |
| | 31–50 | 21 | 9.1 |
| | Over 50 | 3 | 1.3 |
| | High school graduate or below | 4 | 1.7 |
| Education | Bachelor's degree | 158 | 68.1 |
| | Master's degree or above | 70 | 30.2 |
| | Less than 4500 RMB | 193 | 83.2 |
| Income | 4500–7999 RMB | 28 | 12.1 |
| | 8000 RMB or more | 11 | 4.7 |

## 5. Results and Discussion

### 5.1. Common Method Biases Test

Common method biases (CMB) refer to the extent to which the relationship between two variables deviates from the "true score correlation" and results in spurious correlations when both variables are measured by the same respondents [74]. The self-report survey method may raise the common method biases [75]. This study used the method of "controlling the effects of a single unmeasured latent method factor" to test the common method biases, which is suggested by [75]. The brief description of this method is as follows. First, we developed a confirmatory factor analysis (CFA) model, denoted as M1 (RMSEA = 0.047, GFI = 0.971, TLI = 0.964). Subsequently, we extended this model to create M2 (RMSEA = 0.049, GFI = 0.969, TLI = 0.962) by introducing a latent method factor that encompassed all indicators from the other latent factors in the CFA model. Ultimately, if there is no significant difference between these two models, it indicates the absence of common method biases. As shown in Table 3, the difference between the main fitting indexes of M1 and M2 was very small($\Delta$RMSEA = $-0.002$, $\Delta$CFI = 0.002, $\Delta$TLI = 0.002), indicating that there are no obvious common method biases in the measurement model of this study [76].

**Table 3.** Common method biases test.

| Fit Index | M1 | M2 | \|M1 − M2\| |
|---|---|---|---|
| $\chi^2/\mathrm{df}$ [1] | 1.520 | 1.551 | 0.031 |
| RMSEA [2] | 0.047 | 0.049 | 0.002 |
| GFI [3] | 0.971 | 0.969 | 0.002 |
| TLI [4] | 0.964 | 0.962 | 0.002 |

[1] Chi-squared per degree of freedom. [2] root mean square error of approximation. [3] goodness of fit index. [4] Tucker–Lewis Index.

### 5.2. Measurement Model

This study conducted a CFA with MPLUS7.0 to evaluate the measurement reliability and validity. The CFA outcomes revealed that the data fit well with the model ($\chi^2/\mathrm{df}$ = 1.52 < 5, RMSEA = 0.047 < 0.08, GFI = 0.971 > 0.9, TLI = 0.964 > 0.9) [77].

Then, the reliability and validity of the model were measured. First, we evaluated item reliability by calculating the standardized loading of each item on its corresponding construct. Table 4 shows that each item-to-construct loading is larger than the criterion of 0.55. Therefore, all the items are sufficiently reliable [78]. Second, we calculated the composite reliability (CR) of each construct to examine scale reliability. The minimum value

of CR is 0.813 (Table 4), exceeding 0.60, indicating that the scale is internally consistent and reliable [79]. Then, we evaluated the convergent validity by computing the average variance extracted (AVE) value of each construct. All AVE values exceed the criterion of 0.50 (Table 4). Thus, the measurement model has suitable convergent validity [80]. Finally, we compared the square roots of AVE values for all the constructs with their corresponding correlation coefficients with other constructs to evaluate the constructs' discriminant validity. Table 5 reveals that the square roots of AVE values for all the constructs on the diagonal are larger than their corresponding correlation coefficients with other constructs, indicating that the measurement model fulfills the requirements of discriminant validity [81].

**Table 4.** Reliability and convergent validity.

| Construct | Item | Standard Loading | AVE | CR |
|---|---|---|---|---|
| TP | TP1<br>TP2<br>TP3 | 0.875<br>0.798<br>0.623 | 0.597 | 0.813 |
| AL | AL1<br>AL2<br>AL3<br>AL4 | 0.774<br>0.810<br>0.776<br>0.695 | 0.586 | 0.849 |
| PLE | PLE1<br>PLE2<br>PLE3 | 0.863<br>0.918<br>0.803 | 0.744 | 0.897 |
| PPSB | PPSB1<br>PPSB2<br>PPSB3 | 0.755<br>0.880<br>0.869 | 0.700 | 0.874 |
| PP | PP1<br>PP2<br>PP3 | 0.881<br>0.895<br>0.863 | 0.774 | 0.911 |
| GVI | GVI1<br>GVI2 | 0.917<br>0.931 | 0.854 | 0.921 |
| PPSU | PPSU1<br>PPSU2<br>PPSU3 | 0.812<br>0.823<br>0.919 | 0.727 | 0.889 |
| PSI | PSI1<br>PSI2<br>PSI3 | 0.916<br>0.899<br>0.831 | 0.779 | 0.914 |

**Table 5.** Discriminant validity.

| Construct | Mean | St. Dev. | TP | AL | PLE | PPSB | PP | GVI | PPSU |
|---|---|---|---|---|---|---|---|---|---|
| TP | 4.391 | 1.098 | **0.773** | | | | | | |
| AL | 5.240 | 0.988 | 0.469 | **0.766** | | | | | |
| PLE | 3.745 | 0.888 | 0.151 | 0.219 | **0.863** | | | | |
| PPSB | 5.534 | 1.054 | 0.235 | 0.407 | 0.123 | **0.837** | | | |
| PP | 5.083 | 1.077 | 0.406 | 0.471 | −0.045 | 0.579 | **0.880** | | |
| GVI | 5.317 | 1.075 | 0.218 | 0.360 | 0.144 | 0.496 | 0.403 | **0.924** | |
| PPSU | 4.250 | 0.743 | 0.319 | 0.469 | 0.089 | 0.601 | 0.512 | 0.571 | **0.853** |
| PSI | 5.629 | 0.846 | 0.320 | 0.491 | 0.106 | 0.652 | 0.605 | 0.624 | 0.666 |

Note: The bold data on the diagonal represents the arithmetic square root of AVE values, while the data below the diagonal represents the correlation coefficients between variables.

### 5.3. Structural Model

This study adopted the structural equation modeling technique to test the proposed structural model with MPLUS7.0. The fit indices for the structural model are acceptable ($\chi^2$/df = 1.646 < 5, RMSEA = 0.053 < 0.08, GFI = 0.938 > 0.9, TLI = 0.929 > 0.9).

Figure 2 and Table 6 show the results of SEM analysis. First, potential loss expectations have a significant negative effect on perceived privacy ($\beta = -0.125$, $p < 0.05$), and perceived personalized service benefits have a significant positive effect on perceived privacy ($\beta = 0.688$, $p < 0.001$). Therefore, H1 and H2 are supported. Second, perceived privacy ($\beta = 0.181$, $p < 0.001$), group value identification ($\beta = 0.158$, $p < 0.01$), and perceived public service utility ($\beta = 0.645$, $p < 0.0001$) positively affected privacy sharing intentions, confirming H3, H4, and H5, respectively.

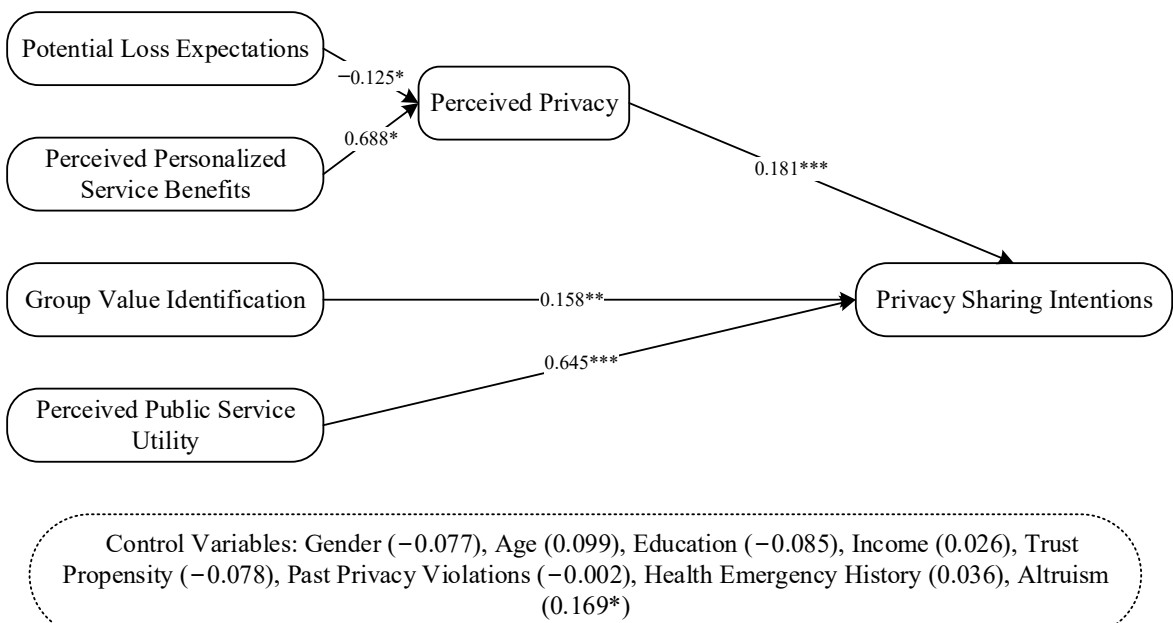

**Figure 2.** Results of SEM analysis. Note: Standardized estimates, * $p < 0.05$, ** $p < 0.01$, *** $p < 0.001$ (two-tailed).

**Table 6.** Path analysis.

|  |  | Estimate | S.E. | C.R. | *p*-Value | Result |
|---|---|---|---|---|---|---|
| H1 | PLE--->PP | −0.125 | 0.058 | −2.144 | 0.032 * | Yes |
| H2 | PPSB--->PP | 0.688 | 0.044 | 15.578 | 0.000 *** | Yes |
| H3 | PP--->PSI | 0.181 | 0.052 | 3.491 | 0.000 *** | Yes |
| H4 | GVI--->PSI | 0.158 | 0.054 | 2.911 | 0.004 ** | Yes |
| H5 | PPSU--->PSI | 0.645 | 0.058 | 11.058 | 0.000 *** | Yes |

Note: Standardized estimates, * $p < 0.05$, ** $p < 0.01$, *** $p < 0.001$ (two-tailed).

### 5.4. The Moderation Effect of Information Type Sensitivity

This study performed a multi-group analysis to test the moderation effect of information type sensitivity with MPLUS7.0 [82]. The samples were divided into two groups according to the type of information collected: a group with low-sensitivity information ($n = 136$) and a group with high-sensitivity information ($n = 96$).

The analysis results are shown in Table 7. The path coefficient of perceived privacy on privacy sharing intentions is 0.054 for the group with low sensitivity information and 0.325 for the group with high sensitivity information. There is a significant difference between these two groups ($\beta = 0.271$, $p < 0.05$). Therefore, information type sensitivity positively moderates the relationship between perceived privacy and privacy sharing intentions,

and H6 is supported. The path coefficient of perceived public service utility on privacy sharing intentions in the low and high sensitivity information context are 0.818 and 0.456, respectively. There is a significant difference between these two groups ($\beta = -0.362$, $p < 0.05$). Therefore, the negative moderation of information type sensitivity on the relationship between perceived public service utility and privacy sharing intentions is proved, and H8 is supported. In contrast, the difference in the path coefficient of group value identification and privacy sharing intentions is not significant ($\beta = 0.177$, $p > 0.05$), indicating H7 is not supported. This reflects the importance of individual privacy rights. Individuals attach great importance to the protection of personal information, especially for sensitive information. Information type sensitivity refers to the sensitivity of individuals to specific information. The more sensitive information is, the more individuals tend to protect their privacy rights. Therefore, when individuals are more sensitive to information types, they will not reduce their privacy sharing intentions due to group value identification.

**Table 7.** Moderation effect of information type sensitivity.

| | | Group with Low Sensitivity Information ($n = 136$) | Group with High Sensitivity Information ($n = 96$) | Difference | Result |
|---|---|---|---|---|---|
| H6 | PLE--->PP | 0.054 | 0.325 | 0.271 * | Yes |
| H7 | GVI--->PSI | 0.072 | 0.249 | 0.177 | No |
| H8 | PPSU---> PSI | 0.818 | 0.456 | −0.362 * | Yes |

Note: Standardized estimates, * $p < 0.05$ (two-tailed).

*5.5. The Mediation Effect of Perceived Privacy*

This study used the bootstrap methodology to check if the impact of potential loss expectations and perceived personalized service benefits on privacy sharing intentions was mediated by perceived privacy. The bootstrap methodology tests the indirect effect by estimating the confidence interval of the indirect effect via bootstrapping. When the confidence interval does not include zero, the indirect effect is significant. As the suggestion of [83], we set the resampling times to 5000. The bootstrapping results show that potential loss expectations (Effect = −0.110, S.E. = 0.056, 95% CI = −0.224~−0.008) and perceived personalized service benefits (Effect = 0.219, S.E. = 0.053, 95% CI = 0.123~0.332) have significant indirect effects (via perceived privacy) on privacy sharing intentions, confirming the mediation effects of perceived privacy on the relationship between potential loss expectations, perceived personalized service benefits, and privacy sharing intentions.

## 6. Conclusions and Implications
### 6.1. Conclusions of Research Findings

The health data tracking system monitors health conditions, behavioral patterns, and health risks by collecting and analyzing individual health data. It also provides personalized health advice and interventions [1]. By analyzing users' health data, businesses gain valuable insights to optimize products and services, offering more personalized and valuable health solutions. Additionally, the utilization of user privacy data within the system benefits both businesses and public services. Via data sharing, companies can contribute to public health policy-making, working together to improve public health [7].

To strike a balance between individual privacy protection and public services, traditional privacy calculus theories need to delve into aspects such as personalized services, social identity, and public interests. This entails considering individual and societal factors and exploring the influencing factors on individuals' privacy cognition, assessment, and decision-making. Thus, this study investigates the impact mechanisms of potential loss expectations, perceived personalized service benefits, group value identification, and perceived public service utility on privacy sharing intentions. By exploring how potential loss expectations and perceived personalized service benefits influence perceived privacy and how perceived privacy affects privacy sharing intentions, a comprehensive research

approach is provided for the development of privacy calculus, offering new perspectives and guidance for the innovation of privacy calculus theories.

Furthermore, the dynamics and differences in individual privacy sharing decisions are aspects that traditional privacy calculus theories have not accurately revealed. Therefore, this study examines the moderating role of information type sensitivity in the relationships between perceived privacy, group value identification, perceived public service utility, and privacy sharing intentions. These studies offer new perspectives to better understand the influence of different information types, group value identification, and perceived public service utility on individuals' privacy sharing intentions. Moreover, they provide an innovative theoretical foundation for designing personalized privacy protection strategies and formulating privacy policies. Via these research findings, a more comprehensive and balanced understanding of users' demands for high-quality services provided by the health data tracking system and the mechanisms influencing their privacy sharing intentions can be achieved, thus promoting further development in system design and sustainable management.

### 6.2. Theoretical Implications and Discussion

6.2.1. Theoretical Implications

This study introduces important assumptions concerning privacy calculus theory and personalized services, offering new perspectives and addressing research gaps in the field. The theoretical implications for each hypothesis are presented as follows.

(1) Theoretical implications of H1 and H2. H1 and H2 fill gaps in privacy calculus theory by considering users' potential loss expectations and perceived benefits of personalized services as influential factors shaping their privacy perceptions in health data tracking systems. These insights enrich the theory and shed light on the complex decision-making process surrounding privacy when personalized and public services coexist. While previous literature has explored individual adoption factors and decision-making processes regarding specific technologies and mentioned risk and benefit assessment by individuals [84], this study offers fresh viewpoints focused on the relationship between personalized service benefits, potential loss expectations, and individual privacy perceptions. These findings provide valuable guidance for a deeper understanding of how personalized services impact individual privacy protection and are crucial in refining privacy protection strategies to achieve a win-win situation for personalized services and privacy protection.

(2) Theoretical implications of H3. H3 highlights the trade-off between personalized services and privacy protection, revealing a positive relationship between individuals' perceptions of privacy protection and their intention to share data. While existing research has explored individuals' privacy perception and privacy management, using various theoretical frameworks and models to explain the relationship between privacy perception and behavior [85], this study specifically focuses on the trade-off between personalized services and privacy protection. This finding provides important clues for understanding the mechanisms of balancing personalized services and privacy, enabling the development of effective privacy protection strategies and achieving a win-win situation between data sharing and privacy protection.

(3) Theoretical implications of H4. H4 emphasizes the positive relationship between group value identification and individual privacy sharing intentions, offering a new theoretical perspective for making trade-off decisions between personalized services and privacy protection. Previous research has focused on the relationship between individuals and groups, as well as trade-off decisions between personalized services and privacy protection. It has highlighted individuals' privacy awareness and concerns and explored how to represent individual interests better in information sharing environments [86]. However, this study brings attention to the positive relationship between group value identification and individual privacy sharing intentions, establishing a theoretical foundation for further exploring the relationship between value identification and privacy sharing, contributing

to a more comprehensive understanding of the decision-making process concerning the balance between personalized services and privacy protection.

(4) Theoretical implications of H5. H5 emphasizes the positive relationship between the perceived public service utility and individuals' privacy sharing intentions. Research indicates that individuals' identification with public services closely relates to their privacy sharing intentions when they perceive positive impacts from these services. While existing studies have focused on the relationship between individuals (such as perceived service utility, privacy sharing intentions, consumer trust, and loyalty) and individual behavior [87], this study emphasizes the trade-off between personalized services and privacy protection from the perspective of perceived public service utility, providing a new angle for decision-making.

(5) Theoretical implications of H6 and H8. H6 suggests that the sensitivity of personal health data plays a positive moderating role in individuals' perception of privacy protection and privacy sharing intentions. This finding highlights the significant impact of personal information sensitivity on shaping individuals' attitudes and behaviors toward privacy protection [88]. Additionally, H8 indicates that the information type sensitivity of personal health data negatively moderates the effect of perceived public service utility on individuals' privacy sharing intentions. While previous research has explored the moderating role of individual information sensitivity in relation to information sharing intentions and considered various influencing factors [89], this study further expands research into the domain of personal health data and the impact of perceived public service utility, providing new insights into individual privacy protection behavior. These findings offer valuable insights for understanding individual privacy protection behavior and exploring the interaction between individual information sensitivity and other factors.

In summary, these hypotheses extend and complement the theory of privacy calculus, offering crucial theoretical support for trade-off decisions between personalized services and privacy protection. Via in-depth research of these assumptions, we can formulate better privacy protection strategies and promote a win-win situation for data sharing and personalized services.

### 6.2.2. Practical Implications

This study offers valuable insights for system design and privacy education, emphasizing the need to strengthen privacy protection and optimize personalized services to enhance individuals' trust in the system and their perception of privacy. To increase data sharing intention, we recommend fostering a sense of identification with collective values among individuals. Simultaneously, individuals' awareness of the public service utility should be enhanced while safeguarding their privacy rights, leading to a win-win situation for data sharing and public services. For health data tracking systems, special attention should be given to personal information sensitivity and different information types should be considered in developing privacy protection measures and effective communication strategies. The specific recommendations are as follows.

(1) Practical implications of H1 and H2. H1 and H2 are particularly valuable for system design and privacy education. System designers can address individual concerns about privacy loss and optimize personalized services by emphasizing privacy protection measures, achieving a balance between personalized services and privacy protection. Individual privacy education and awareness are also critical. Strengthening privacy education and increasing transparency in data usage help individuals better understand the value of their personal data and privacy rights, empowering them to make informed decisions about privacy protection.

(2) Practical implications of H3. H3 measures can be taken in system design, personal privacy education, and data sharing decisions. Privacy protection measures in personalized services reduce individual concerns about privacy loss and improve the provision of personalized services. Transparent data usage purposes increase trust in data sharing. Emphasizing the importance and legitimacy of data sharing in personal privacy education

fosters a better understanding of the data sharing process and increases confidence in data sharing. Adapting the degree of personalized services based on individuals' perception of privacy protection enables more informed data sharing decisions and better protection of privacy rights.

(3) Practical implications of H4. According to H4, by enhancing individuals' and communities' sense of shared values, individuals become more willing to share data. Integrating personalized services with public services and emphasizing individual participation in supporting collective values can be effective. Strengthening group value identification also plays a key role in encouraging data sharing while safeguarding privacy rights.

(4) Practical implications of H5. H5 is of practical significance, which guides system design, personal privacy education, and data sharing policy formulation. Managers can introduce incentive mechanisms to encourage active user participation in data sharing, which may involve acknowledging their involvement or providing feedback to demonstrate the value of their data in enhancing public services.

(5) Practical implications of H6 and H8. From a practical perspective, H6 and H8 hold great significance for health data tracking systems involving personalized and public services. Personal information sensitivity significantly impacts privacy protection. Privacy protection measures must consider personal information sensitivity to increase users' data sharing intention. Furthermore, the health data tracking system should consider not only the public service utility but also the information type sensitivity of individuals. System administrators can provide users with personalized privacy settings to align with their varying levels of information type sensitivity. This empowers users to independently select the extent to which they wish to share information based on their privacy preferences, thereby enhancing their perception of control and increasing their willingness to share.

*6.3. Limitations and Future Research*

This study does not consider privacy management mechanisms, as well as information sharing and collaboration mechanisms between users and enterprises. Future research will consider privacy management mechanisms, including users' control over data, selective sharing, and recycling rights, to improve users' trust and security in the health data tracking system. In addition, research will be conducted on users' data sharing intention between personalized and public services, and corresponding information sharing and collaboration models will be designed. This can include cooperation models between users, health institutions, researchers, etc., to achieve a win-win situation between personalized services and public services. Finally, differential privacy protection mechanisms can also be considered. Investigating the application of differential privacy technology in the health data tracking system of personalized services and public services. Differential privacy protects the privacy of personal data by introducing certain noise or randomization, thus balancing the relationship between personalized services and privacy protection.

**Author Contributions:** Conceptualization, S.L. and B.Z. (Beiyan Zhang); methodology, B.Z. (Beiyan Zhang); software, B.Z. (Beiyan Zhang); validation, K.P., H.C. and R.L.; formal analysis, B.Z. (Beiyan Zhang); investigation, B.Z. (Beiyan Zhang); resources, H.C. and R.L.; data curation, K.P.; writing—original draft preparation, B.Z. (Beiyan Zhang); writing—review and editing, H.C., R.L., B.Z. (Boyi Zhu) and Z.L.; visualization, H.C., R.L., B.Z. (Boyi Zhu) and Z.L.; supervision, S.L.; project administration, S.L.; funding acquisition, S.L. All authors have read and agreed to the published version of the manuscript.

**Funding:** This research was supported by the National Natural Science Foundation of China (No. 71871135 and No. 72271155).

**Institutional Review Board Statement:** Not applicable.

**Informed Consent Statement:** Informed consent was obtained from all subjects involved in the study.

**Data Availability Statement:** The data presented in this study are available upon reasonable request from the corresponding author.

**Conflicts of Interest:** The authors declare no conflict of interest.

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
