# Peer review of "Research on Users’ Privacy-Sharing Intentions in the Health Data Tracking System Providing Personalized Services and Public Services"

_sustainability, doi:10.3390/su152215709_

Round 1
Reviewer 1 Report
Comments and Suggestions for Authors
Dear authors,
Thank you for the opportunity to review your paper. The study aimed to examine the influence of risk perception and factors in the context of personalized and public services.
Overall, the Introduction is well-written. However, the last 6 paragraphs are confusing to the reader. The purposes of the studies are repeated and described several times. Finally, the results are described in the last paragraph. This does not make sense to me in an Introduction section.
The paper has a deep background regarding the literature on the topic, however, the structure adopted is confusing to me. This seems to be a research article, and the literature review section should be included in the Introduction body. The hypothesis should also be included in the Introduction together with the study’s purposes.
I think this is an interesting paper and topic, but I believe the paper needs to be reformulated before it can be considered for publication. I recommend the authors to simplify the information presented in the first section of the paper (literature), focusing more in their results and discussion.
Reviewer 2 Report
Comments and Suggestions for Authors
The present study addressed the issue of using privacy data from users of the health data tracking system. In general, the authors printed a good theoretical framework in the introduction section, followed by a clear structuring and presentation of hypotheses. Therefore, methodological procedures, including analyzes were well structured, offering interesting results to the academic community.
I have a small suggestion:
1. Considering that the authors presented and discussed the data at the same time. It would be interesting to title section number 5 Results and Discussion.
Reviewer 3 Report
Comments and Suggestions for Authors
The article delves into the utilization of user privacy data within a health data tracking system (HDTS) for both business and public service purposes. It successfully identifies gaps in previous research, with a particular focus on the factors influencing users' intentions to share their private data. Notably, it establishes positive relationships between perceptions of privacy protection, group value identification, and users' willingness to share their data.
The paper is commendably well-written and organized. I recommend it for publication with minor revisions:
- Consider relocating Lines 151-166 from the introduction to a more suitable section, as they appear to be more in line with the study's outcomes and recommendations.
- It would be beneficial to include a brief description of the Common Method Biases test used in Section 5.1 to enhance clarity and understanding.
- The paper would benefit from a dedicated discussion section. To improve clarity, I suggest dividing the current conclusion section into two parts: "Discussion" and "Conclusion."
- Conclusions should be presented in a concise bullet-point format for clarity and easy reference.
Minor editing may be required.
Round 2
Reviewer 1 Report
Comments and Suggestions for Authors
Dear authors,
I appreciate your effort in improving this manuscript and I accept it in the current form.